# In Situ DNA/Protein Interaction Assay to Visualize Transcriptional Factor Activation

**DOI:** 10.3390/mps3040080

**Published:** 2020-11-21

**Authors:** Michela Corsini, Emanuela Moroni, Cosetta Ravelli, Elisabetta Grillo, Marco Presta, Stefania Mitola

**Affiliations:** Department of Molecular and Translational Medicine, University of Brescia, 25123 Brescia, Italy; manumoroni@yahoo.it (E.M.); cosetta.ravelli@unibs.it (C.R.); elisabetta.grillo@unibs.it (E.G.); marco.presta@unibs.it (M.P.)

**Keywords:** transcriptional factor, chick embryo CAM, CREB, DNA/protein interaction

## Abstract

The chick embryo chorioallantoic membrane (CAM) represents a powerful in vivo model to study several physiological and pathological processes including inflammation and tumor progression. Nevertheless, the possibility of deepening the molecular processes in the CAM system is biased by the absence/scarcity of chemical and biological reagents, designed explicitly for avian species. This is particularly true for transcriptional factors, proteinaceous molecules that regulate various cellular responses, including proliferation, survival, and differentiation. Here, we propose a detailed antibody-independent protocol to visualize the activation and nuclear translocation of transcriptional factors in cells or in tissues of different animal species. As a proof of concept, DNA/cAMP response element-binding protein (CREB) interaction was characterized on the CAM tissue using oligonucleotides containing the palindromic binding sequence of CREB. Scrambled oligonucleotides were used as controls. In situ DNA/protein interaction protocol is a versatile method that is useful for the study of transcription factors in the cell and tissue of different origins.

## 1. Introduction

The chick embryo chorioallantoic membrane (CAM) is an extraembryonic membrane that exhibits an extensive and well-organized capillary network composed of narrow arteries and veins and of lymphatic vessels that envelops the chick embryo [1,2,3,4]. The development of the CAM starts from day 3 post-incubation [5]. Blood vessels proliferate until day 11. After that time, endothelial mitotic index declines, and the vasculature system attains its final arrangement at day 18 [6]. The CAM represents a well-established model to investigate several physiological and pathological processes including angiogenesis, tumor growth and dissemination [3,7,8,9], gas exchange [10], and ion transport [11]. The CAM has been also exploited to study material biocompatibility [12], selective vascular occlusion therapies [13], drug distribution, and to perform toxicological analyses [14,15]. The easy access for experimental manipulation and the low cost make this system a valid alternative to other in vivo models [16,17]. Despite these advantages, there are also some limitations in the use of CAM in experimentation, especially for antibody-based assays. Only a few antibodies have been developed, or at least tested, against avian antigens. For instance, tumor-infiltrating neovessels and the associated mural cells cannot be easily characterized (analyzed or visualized) by specific anti-chicken antibodies. Similarly, the characterization of phosphorylation and activation of second messengers or transcription factors can be particularly challenging. Among the transcriptional factors, cAMP response element-binding protein (CREB) is a 37 KDa downstream effector of a wide plethora of extracellular and intracellular stimuli that are essential to control cell functions both under physiological and pathological circumstances [18,19]. CREB belongs to a large family of basic leucine zipper (bZIP)-containing transcription factors, including c-jun, c-fos, and c-myc (10.1073_pnas.0501076102) [20]. Reversible phosphorylation on serine 133 and 142 modulates CREB activation, nuclear translocation, and DNA interaction. Sumoylation of Lys304 is required for nuclear localization of CREB [21]. CREB transcription can be enhanced by TORC coactivators, which act independently of Ser-133 phosphorylation [22,23]. Activated CREB binds a highly conserved octameric palindromic DNA sequence (5′-TGACGTCA-3′), named cAMP response elements (CRE), or a slight variant that are located within the promoter or enhancer regions of several viral and cellular genes [24]. Several antibody-based approaches have been set up to assess CREB/DNA interaction, including the electrophoretic mobility shift assay (EMSA) combined with the supershift assay, the chromatin immunoprecipitation (ChIP) assay, and ELISA [25].

Here, we propose a detailed oligonucleotide-based protocol to study CREB activation directly in avian tissues. This in situ DNA/protein interaction assay is suitable for the visualization of the spatial–temporal activation of transcriptional factors other than CREB and can be applied to cells and tissues of various animal species in different experimental models.

## 2. Experimental Design

### 2.1. Materials

#### 2.1.1. Probes

Biotin or Cy5-labeled CRE oligo-probe forward 5′-AGAGATTGCCTGACGTCAGAGAGCTAG -3′ (Sigma, Merck group).Biotin or Cy5-labeled CRE oligo-probe reverse 5′-CTAGCTCTCTGACGTCAGGCAATCTCT-3′ (Sigma, Merck group).Biotin or Cy5-labeled CRE scramble oligo-probe sense 5′-AGACATTGCCTGGATAGGGAGAGTTAG-3′ (Sigma, Merck group).Biotin or Cy5-labeled CRE scramble oligo-probe nonsense 5′-CTAACTCTCCCTATCCAGGCAATGTCT-3′ (Sigma, Merck group).

#### 2.1.2. Reagents

Alginic Acid Sodium Salt powder (Sigma, Merck Group–cat n° A2158).Calcium Chloride dihydrate (Sigma, Merck Group–cat n° C3306).PBS (Dulbecco’s Phosphate-buffered saline w/o Ca and Mg, 10×; Lonza–cat n° BE17–517Q). Dilute 1:10 for working solution.Triton X-100 (Sigma, Merck Group–cat n° T8787). **! CAUTION** Triton X-100 is hazardous. Avoid contact with skin and eyes.Killik (Optimal cutting temperature compound (OCT); Bio-Optica–cat n° 05-9801).Salmon Sperm DNA (ThermoFisher-cat n° AM9680).Avidin Blocking Reagent (Vector Laboratories Inc.–cat n° SP-2001).Methanol (Sigma, Merck Group–cat n° 32213). **! CAUTION** Methanol is toxic. Manipulate in a fume hood.Streptavidin AlexaFluor 488 or 594 (Molecular Probes–cat n° S11223–S11227).DAPI (4′,6-diamidin-2-fenilindolo) (Sigma, Merck Group–cat n° D8412). Dilute 1:15000 for working solution.Dako Fluorescence Mounting Medium (Dako Cytomation–cat n° S3023).

#### 2.1.3. Eggs

Fertilized white leghorn eggs.

### 2.2. Equipment

Static incubator for eggs (FIEM snc, Buttigliera, Italy).Surgery straight forceps (2Biological Instrument–cat n° 11255-20).Surgery straight scissors (2Biological Instrument–cat n° 14094-11).Surgery curved scissors (2Biological Instrument–cat n° 14095-11).5 ml syringe (Terumo–cat n° SS*05SE1).Leukosilk Silk Tape 2.5 cm × 5 m (BSN Medical–cat n° 01022-00).Transparent tape.Cryostats.Fluorescence Microscope and/or Confocal Microscope.

## 3. Procedure

### 3.1. Egg Preparation

The following procedure is for “in ovo” experiments. 

Gently wash the eggs in 25–28 °C water, and incubate sagittally at 37 °C in a humidified incubator.Four days post-incubation, prick the large end of the shell in correspondence with the air space with a forceps (Figure 1–STEP 1). This procedure allows us to break the inner shell membrane and detach the embryo from the eggshell.Drill a hole on the opposite side to the first one, and aspirate 3 to 5 mL of egg albumen with a syringe without a needle (Figure 1–STEP 2).Promptly close the holes with silk tape. Incubate the eggs at 37 °C for 24 h.Place a piece of transparent tape on the eggshell, and make it adhere perfectly.Cut a window in the eggshell approximately 1.5 × 2.5 cm wide with a curve surgical scissor (Figure 1–STEP 3 and STEP 4).Close the egg with transparent tape (Figure 1–STEP 5). **CRITICAL STEP**: ensure that the tape is well tightened to prevent the embryo from drying out.Incubate the eggs at 37 °C for 7 d.

#### 3.1.1. Egg Treatment

Several engraftment methods to convey molecules, cells, or tissues on chick embryo CAM have been described. Among them, here, we use a hydrogel of calcium alginate combined with the molecules under testing.

Prepare a solution of Alginic acid 6% (*w*/*v*) in sterile milliQ H_2_O at least 16–18 h before starting.Spot a drop (3–4 μL) of Alginic acid solution into the lid of a culture dish and immediately add 3 uL of the molecule under testing.Add a drop of CaCl2 0.1 M to allow the polymerization of Alginic acid.Gently transfer the alginate pellets on top of the CAM using a forcep.

#### 3.1.2. Sample Harvest and Inclusion

Following the timescale provided by your protocol, collect the samples without fixing them.Gently wash the CAM with cold PBS for 2–3 min.Include the CAM into tissue embedding resin OCT for cryosections and immediately submerge in liquid nitrogen for snap freezing at least 30 sec. **! CAUTION** Extremely cold liquid (−196 °C) can cause severe frostbite and cold burns. Gloves and face shield required.

**PAUSE POINT**. At this stage, the samples can be stored at −80 °C (for over 1 year) or −20 °C (for 6 to 8 months).

Cut 4–5 μm thick sections, and store the samples at −80 °C.

**PAUSE POINT**. At this stage, the samples can be stored at −80 °C (for over 1 year) or −20 °C (for 6 to 8 months).

#### 3.1.3. Hybridization and Staining

Wash tissue sections in phosphate buffer saline (PBS) 1% Triton X-100 for 2–3 min.Rinse them in PBS for 5 min.Incubate with Avidin blocking solution 1:20 in PBS for 1 h at room temperature (RT).Rinse in PBS for 5 min.Incubate with Salmon sperm DNA 100 ug/mL in PBS for 1h at RT.Add biotinylated CREB oligos 600 pmol/reaction and incubate 45 min at RT.Rinse twice with PBS for 5 min.Fix with cold methanol for 10 min.Rinse twice with PBS for 5 min.Add AlexaFluor 488-conjugated Streptavidin or AlexaFluor 594-conjugated Streptavidin 10 μg/mL in PBS for 1 h at RT in a dark humidified chamber.Rinse twice with PBS for 5 min.In the dark, counterstain nuclei with DAPI 50 ng/mL in PBS for 15 min at RT.Mount the slices with Dako Mounting Medium and an appropriate coverslip.

**PAUSE POINT**. At this stage, the sections can be stored at 4 °C for a couple of months.

## 4. Expected Results

Here, we developed an oligonucleotide-based in situ assay to characterize the activation of transcriptional factors in whole tissues of different animal species. The Activating transcription factors (ATF)/CREB protein family underwent multiple duplication events during the evolution process, generating heterogeneous homologous proteins. Thus, all the antibody-based assays aimed to investigate the spatial-temporal activation of CREB require protein-specific as well as species-specific antibodies. The protein alignment of the sequence of the ATF/CREB family members of different species indicates that the DNA binding domain, unlike the other regions of the protein, has been highly conserved during the evolution of animal genomes (Figure 2). To directly analyze the CREB/DNA interaction in samples of different species, we propose the use of labelled oligonucleotides based on the palindromic CREB binding sequence as an alternative probe to the use of anti-CREB antibodies.

Similar to other transcription factors, CREB activity is exerted by binding to a conserved CRE sequence that occurs either as a palindrome (TGACGTCA) or half-site (CGTCA_TGACG) [26]. In our previous paper, we showed that gremlin, a vascular endothelial growth factor receptor 2 (VEGFR2) ligand, leads to endothelial cell proliferation, migration, and survival through CREB activation in vitro [27,28]. Here, we used gremlin to induce the activation and nuclear translocation of CREB in vivo in the chick embryo CAM assay.

To this purpose, CAMs were treated with 100 ng/implant of recombinant mouse gremlin, harvested, washed in PBS, and included in OCT for the snap-freeze. Five micrometer slices were cut using a cryostat. Samples were permeabilized with Triton X-100 and incubated with Salmon Sperm DNA to reduce unspecific interaction. When biotin-labeled oligoprobes were used, incubation with an avidin blocking solution was additionally included to neutralize all biotin binding sites into the sections. Importantly, avidin/biotin interaction is unaffected by extremes in pH, temperature, organic solvents, and other denaturing agents, so we can consider this treatment as irreversible. Then, sections were incubated with the Cy5- or biotin-labeled palindromic CRE oligo-probe or a Cy5- or biotin-labeled CRE scrambled sequences for 45 min. When fluorescent-oligo-probes were used, samples were washed and directly mounted in Dako Mounting Medium. When sections were incubated with biotin oligoprobes, samples were incubated with fluorescent-conjugated Streptavidin for 1 h before mounting. CAM samples were analyzed using an epifluorescent microscope. As anticipated, gremlin induces the activation and nuclear translocation of CREB in endothelial cells of the CAM, as demonstrated by the presence of fluorescent positive nuclei in samples incubated with the consensus CRE oligo (Figure 3). No positive nuclei are present in CAMs treated with vehicle (PBS). The absence of fluorescent positive cells in gremlin-stimulated CAMs incubated with the scrambled oligo-probe points out the specificity of the in situ assay. The in situ DNA/protein interaction protocol was applied to visualize gremlin-induced CREB activation in murine cells in vitro (Figure 4). To this purpose, murine endothelial cells were plated on a glass coverslip, stimulated for 10 min with recombinant gremlin (50 mg/mL), and fixed in 3% paraformaldehyde/2% sucrose in PBS for 30 min. Next, cells were permeabilized with 0.5% Triton-X100 for 10 min to allow the entry of oligo-probes and incubated with Cy5-labeled CRE oligo probe. Of note, the in situ DNA/protein interaction was previously on human endothelial cells with similar results [27]. These data support the idea that the same oligo-probes can be used to visualize the activation of transcriptional factors in various experimental models in different animal species.

The chick embryo CAM is a robust in vivo model to study several biological processes. However, the lack of reagents can limit the possibilities of analysis. The in situ DNA/Protein Interaction Assay can be adapted to study in CAM the effects of soluble molecules as well as cells on other transcriptional factors including NF-kB (Moroni E., unpublished observation). This confers an additional feature to the chick embryo CAM as an experimental platform in medical research for a fine characterization of the intracellular pathways, underlying physiological and pathological processes and the response to therapeutic treatments. Importantly, all the reagents used for in situ DNA/protein interaction are compatible with the immunofluorescence protocols; thus, DNA/protein in situ can be combined with them.

## Figures and Tables

**Figure 1 mps-03-00080-f001:**
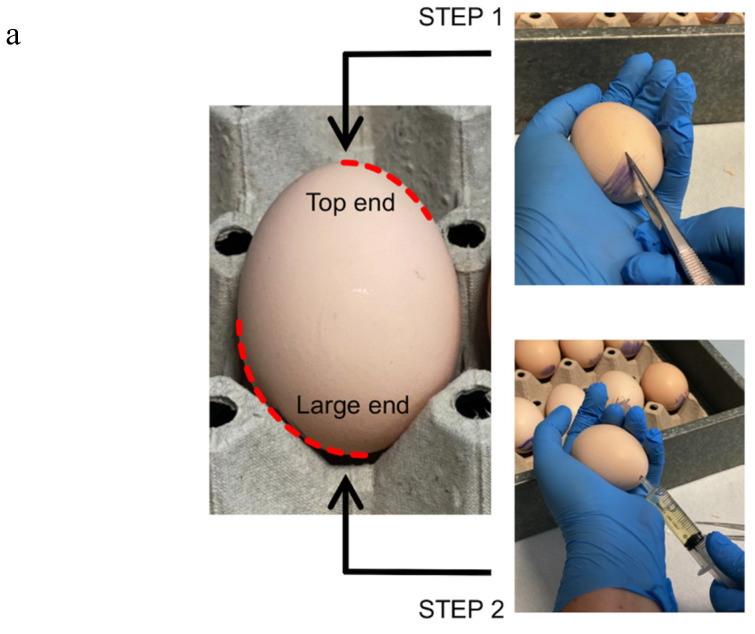
Handling of eggs: (**a**) manipulation of the egg to suck the albumen; (**b**) opening of the shell, exposure of the chorioallantoic membrane (CAM) for the implants, and closing with tape.

**Figure 2 mps-03-00080-f002:**
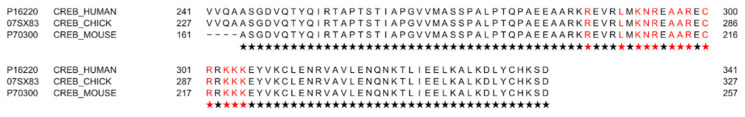
Alignment of cAMP response element-binding protein (CREB) protein sequences from human, mouse, and chicken origin. Stars indicate fully conserved amino acids. The amino acids in red indicate the DNA binding sequence.

**Figure 3 mps-03-00080-f003:**
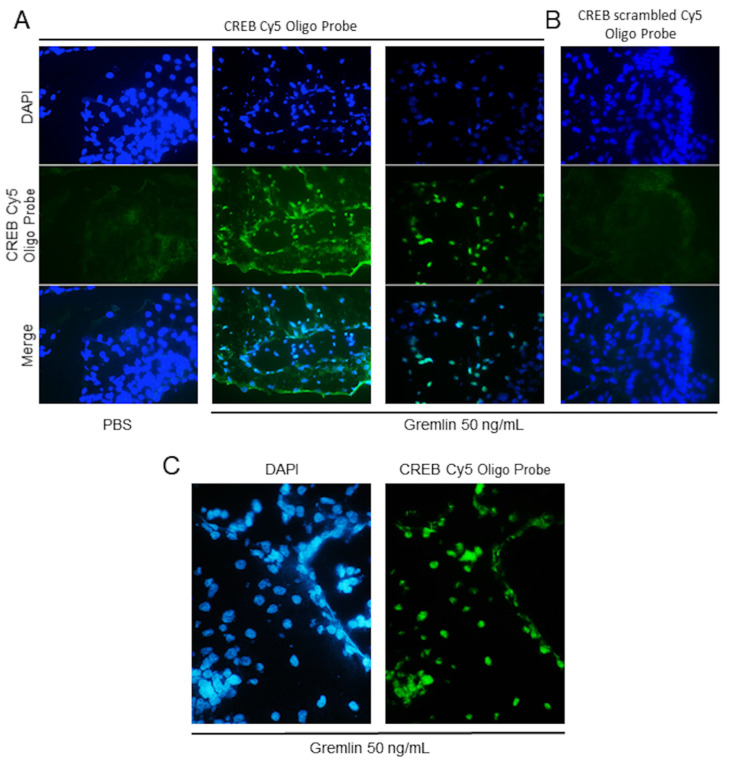
In situ DNA/CREB interaction in CAM sections. (**A**) Control and gremlin-stimulated CAM sections (5 μm) were permeabilized and incubated with Cy5-labeled palindromic CREB binding oligo-probes. Nuclei were counterstained with DAPI (magnification 20×). (**B**) Gremlin-stimulated CAM sections (5 μm) were permeabilized and incubated with Cy5- scrambled oligo-probes. (**C**) Magnification (40×) of gremlin-stimulated CAM sections (5 μm).

**Figure 4 mps-03-00080-f004:**
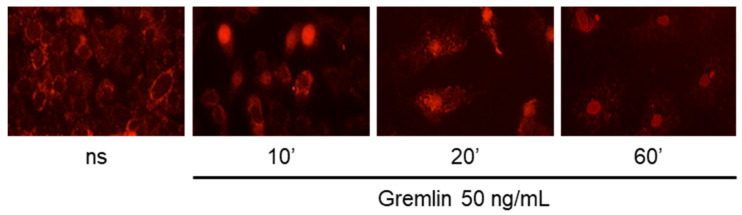
In situ DNA/CREB interaction on murine endothelial cells. Endothelial cells stimulated with murine recombinant gremlin for 0, 10, 20, 60 min, washed and subjected to in situ DNA/CREB–interaction assay. Images were acquired using a Zeiss Axio Imager.A2 microscope equipped with 20× EC Plan-NEOFLUAR objective.

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
