# Peer review of "In Situ DNA/Protein Interaction Assay to Visualize Transcriptional Factor Activation"

_mps, 2020, doi:10.3390/mps3040080_

Round 1

Reviewer 1 Report

In this manuscript, Michela et al presented an antibody-independent protocol to visualize transcriptional factor activation, based on in situ DNA/protein interaction assay.

Specific concerns,

1, neither figure 2 nor figure 3 is cited in the main text.

2,  in figure 3, each panel needs to be labeled.

3, in figure 4, the 60 min image is apparently less bright than that of the 10 min and 20 min, actually it’s as dimmed as the control (ns).

  1. figure 4 did not demonstrate the translocation of the protein, ies there is no image showing that the protein is located in the cytoplasm of the cells.

Author Response

We thank the reviewer for appreciating our work.

 Specific concerns,

1, neither figure 2 nor figure 3 is cited in the main text.

We apologize for the  oversight. We introduced the citation of all figures in the main text. 

2,  in figure 3, each panel needs to be labeled.

We implemented Fig 3 with specific labels.

3, in figure 4, the 60 min image is apparently less bright than that of the 10 min and 20 min, actually it’s as dimmed as the control (ns).

And 4. figure 4 did not demonstrate the translocation of the protein, ies there is no image showing that the protein is located in the cytoplasm of the cells.

 We apologize if this was not clear. We substituted the previous image with a new ones more magnified . In the new figure is easier to appreciate the perinuclear localization of eventually activated CREB proteins in basal condition . When CREB activation was followed by immunofluorescence, pCREB iis clearly localized around nuclei in the first phase of activation  before its tralocation ( see Corsini DOI: 10.1161/ATVBAHA.113.302517)

Reviewer 2 Report

The article "In situ DNA/protein interaction assay to visualize transcriptional factor activation" describes the use of labeled oligonucleotide corresponding to response elements to detect the
location/nuclear translocation of transcription factors, especially for cases where antibodies against the studied transcription factor are not readily available, such as less studied species.
The authors show the applicability of the method in chick embryo chorioallontoic membranes (CAMs), but demonstrate the method can also be used in murine cells.
The article is clear, well written and the method can be very useful o the scientific community. However, the authors only study one transcription factor (CREB) and nevertheless claim that the method will be useful for other transcription factors.

Remarks:

While the authors demonstrate that the method can be used in different cell types of different species, they do not demonstrate that it works for different transcription factors.
To make this more credible, useful and less of a risky adventure for other studies in other research teams, and to improve the chances that the method will be picked up by other teams, it is important to demonstrate that the method can indeed be used for other transcription factors. This can be demonstrated in any cell type.

In this respect, the article "Subcellular localization as a limiting factor for utilization of decoy oligonucleotides" , Nucleic Acids Res. 2004; 32(19): e142.
does mention some possible problems: for some transcription factor such as AP-1 , oligo's do not seem to go to the nucleus at all.

legend to figure 2: asterisks usually indicate identical amino acids rather than conserved ones?

line 208: description of the experiment in CAMs refers to figure 4, but this should probably refer to figure 3?

line 182: all the ab-based assays aimed to investigate spatial-temporal activation of CREB require protein-specific (I believe the - between protein and specific is missing in the MS)
as well as species-specific antibodies. "as well as species-specific abs ": It is unclear why the species specificity is so important here, a it is no problem that the ab would recognize CREB of other species as well.

description in figure 4: no DAPI staining is shown, and their is also very focal staining in the non-stimulated condition that may correspond to nuclear staining.
The DAPI staining should be shown and a blinded experiment should be carried out with blind assessment of the nuclear staining (co-localization of DAPI and CREB specific probe) should be reported.

It is not clear whether this approach is entirely new, and no references to literature for similar approaches are given. references to such a literature study should be added.

Author Response

In this respect, the article "Subcellular localization as a limiting factor for utilization of decoy oligonucleotides" , Nucleic Acids Res. 2004; 32(19): e142.
does mention some possible problems: for some transcription factor such as AP-1 , oligo's do not seem to go to the nucleus at all.

The use of decoy oligonutleotids is usfull in vivo models, where in vivo means all live cell and animal based assays. In these experimental conditions the oligo decoy binds and prevents transcriptional factor nuclear translocation. In our lab during the analysis included in  Corsini ATVB 2014, we also used with the decoy CREB oligo to inhibit creb activity , unfortunately with low efficiency. Thus we changed the inhibitor approches and cells were trasfected cells with dominant-negative bZIP domain mutant ACREB (Hiroki Ono et all Hypertension research 2006) . 

However, the goal of this paper is to described a molecular assay usefull for fixed samples ( cell lines and tissues) where  sample permeabilization allows the nuclear stain.  

legend to figure 2: asterisks usually indicate identical amino acids rather than conserved ones?

We used asterisks to indicate conserved and identical aa

line 208: description of the experiment in CAMs refers to figure 4, but this should probably refer to figure 3?

line 182: all the ab-based assays aimed to investigate spatial-temporal activation of CREB require protein-specific (I believe the - between protein and specific is missing in the MS)

We apologize fot the mistake we modified Fig 4 to Fig 3 and protein-specific to protein specific

as well as species-specific antibodies. "as well as species-specific abs ": It is unclear why the species specificity is so important here, a it is no problem that the ab would recognize CREB of other species as wel

The oligo test overcomes the antibody anavailability for alternative research models inclusing chicken and zebrafish 

description in figure 4: no DAPI staining is shown, and their is also very focal staining in the non-stimulated condition that may correspond to nuclear staining.
The DAPI staining should be shown and a blinded experiment should be carried out with blind assessment of the nuclear staining (co-localization of DAPI and CREB specific probe) should be reported.

We apologize if this was not clear. We substituted the previous image with a new ones more magnified . In the new figure is easier to appreciate the perinuclear localization of eventually activated CREB proteins in basal condition . When CREB activation was followed by immunofluorescence, pCREB iis clearly localized around nuclei in the first phase of activation  before its tralocation ( see Corsini DOI: 10.1161/ATVBAHA.113.302517)

It is not clear whether this approach is entirely new, and no references to literature for similar approaches are given. references to such a literature study should be added.

We previously described the dna/ based assay on human cells  see line 217

Reviewer 3 Report

In this study, the authors are describing the developing protocol for an in situ DNA/protein interaction assay to visualize transcription factor activation. They propose a detailed oligonucleotide-based protocol to study CREB activation directly in the chick embryo chorioallantoic membrane (CAM). This in situ DNA/protein interaction assay is applicable for the visualization of the spatial- temporal activation of transcriptional factors other than CREB and can be applied to cells and tissues of various animal species in different experimental models. This is a well written manuscript to be published once some minor errors listed below are amended appropriately.

  1. Check grammatical errors throughout the manuscript.
  2. Indicate Fig2 and Fig3 in the text.
  3. Fig3: Labeling (A,B) is missing. Besides, the magnified images in Fig3 do not seem to correspond to the unmagnified image (Red rectangle).
  4. Line199: avidin/binding à avidin/biotin

Author Response

We thank the reviewer for appreciating our work. We checked the grammatical errors, included fig 2 and fig 3 in the main text, added the fig 3 labeled , and cleared the red rectangled and inserted a CAM higher magnification in Fig 3